# TOWARDS CHEAPER INFERENCE IN DEEP NETWORKS WITH LOWER BIT-WIDTH ACCUMULATORS

**Yaniv Blumenfeld**
Technion, Israel
yanivblm6@gmail.com

**Itay Hubara**
Intel-HabanaLabs, Israel
itayhubara@gmail.com

**Daniel Soudry**
Technion, Israel
daniel.soudry@gmail.com

## ABSTRACT

The majority of the research on the quantization of Deep Neural Networks (DNNs) is focused on reducing the precision of tensors visible by high-level frameworks (e.g., weights, activations, and gradients). However, current hardware still relies on high-accuracy core operations. Most significant is the operation of accumulating products. This high-precision accumulation operation is gradually becoming the main computational bottleneck. This is because, so far, the usage of low-precision accumulators led to a significant degradation in performance. In this work, we present a simple method to train and fine-tune high-end DNNs, to allow, for the first time, utilization of cheaper, 12-bits accumulators, with no significant degradation in accuracy. Lastly, we show that as we decrease the accumulation precision further, using fine-grained gradient approximations can improve the DNN accuracy.

## 1 INTRODUCTION

Deep Neural Networks (DNNs) quantization (Hubara et al., 2017; Sun et al., 2020; Banner et al., 2018; Nagel et al., 2022; Chmiel et al., 2021) have been generally successful at improving the efficiency of neural networks' computation without harming the accuracy of the network Liang et al. (2021). The suggested methods aim to reduce the cost of the Multiply-And-Accumulate (MAC) operations for both training and inference. To this end, they quantize the weights, activations, and gradients. For applications utilizing such quantization methods, the cost of multiplications, commonly considered to be the computational bottleneck, can be substantially reduced. However, the accumulation of computed products is still performed with high-precision data types. Consequently, the cost of the accumulation, as a component of MAC operations, becomes increasingly dominant in performance breakdowns (Sakr et al., 2019; Ni et al., 2020; Chmiel et al., 2021).

For example, when the weights and activations are in the common FP8 format, van Baalen et al. (2023) showed the accumulation becomes a computational bottleneck. For example, they conducted experiments to estimate the raw gate count for various FP8 implementations (a first-order approximation for power and area) and observed a $2\times$ reduction in gate count when employing FP16 accumulators instead of FP32. Similarly, Ni et al. (2020) reported analogous findings for INT8, demonstrating that an 8-bit×8-bit multiplier consumes a comparable amount of power and silicon area to a 32-bit accumulator.

In this study, we focus on reducing the numerical precision of the accumulation operation in DNNs. Building our solution on top of the emerging FP8 format, which has gained prominence for both training and inference on the most prevalent hardware Andersch et al. (2022), we aim to optimize such DNNs, to enable inference on hardware with Low Bit-width Accumulators (LBAs).

Our main contributions are:

- We propose a simple scheme for fine-tuning models with 12-bit accumulators for a variety of tasks, and show this method can already achieve strong performance. For example, we show for the first time that 12-bits accumulators can be used in ResNets on ImageNet, with no significant degradation in accuracy.
- We examine more fine-grained approaches, in which, for the first time, we backpropagate through the entire accumulation-computation graph. Though much more expensive during training, such fine-grained backpropagation can be used to significantly improve the accuracy of DNNs with LBAs at lower bit-widths.

## 2 PRELIMINARIES: QUANTIZED NEURAL NETWORKS

### 2.1 QUANTIZED WEIGHTS AND ACTIVATIONS

The quantization of neural networks is, by now, a standard practice for achieving efficient neural networks. Unlike traditional scientific computation, that often (Bailey, 2005) requires high-precision floating point arithmetic (e.g., FP64) to achieve accurate results, it was observed (Gupta et al., 2015) that deep neural networks can maintain high accuracy when the weights and activations in the network are represented in low bit representation. As a result, training deep neural networks (DNNs) using half-precision (FP16) arithmetic became the default setup for modern Deep Learning applications (Brown et al., 2020). Lower precision representation (INT8, FP8, INT4, FP4, and Binary) (Sun et al., 2019; 2020; Courbariaux et al., 2016) is also used for a variety of deep learning applications, for either training or inference, albeit using them is more experimental and may result in lower model performance, depending on the specific application.

Quantization of Weights and Activations (W/A) has two main benefits.

- *Lower memory footprint*: By reducing the number of bits used for representation of each numerical value, W/A quantization can significantly reduce the memory required for storing and using a neural network. Consequently, W/A quantization enables storing larger models (with more parameters and activations) on DL accelerators with finite storage and improves the computation efficiency of smaller models by mitigating memory bottlenecks.

- *Reduced complexity of multiplication operation*: Neural networks commonly compute multiplications of weight and activation pairs. When both weight and activation are represented at a lower precision, it is possible to perform the multiplication operation with cheaper hardware (smaller area, less energy). This allows us to do more multiplication operations per second, provided that the hardware was designed to support these lower-precision operations.

Numerical values are typically represented using either fixed, or floating point format. Methods for quantization of DNNs can be divided accordingly.

### 2.2 FIXED POINT QUANTIZATION

Given a full-precision value $x$, a fixed number of bits $B$, and an integer $b$ (exponent-bias), we define the fixed-point quantization of $x$ as:

$$R_{\min} \equiv -2^{B-b-1} \qquad Q_{B,b}^{\text{FIXED}}(x) \equiv \begin{cases} R_{\min} & x \leq R_{\min} \\ R_{\max} & x \geq R_{\max} \\ 2^{-b}\text{Round}\left(x \cdot 2^b\right) & \text{else} \end{cases} \qquad (1)$$
$$R_{\max} \equiv 2^{-b}\left(2^{B-1} - 1\right)$$

As we can see from Eq. (1), the process of fixed-point quantization involves two explicit changes to the value of $x$. First, we round $x \cdot 2^b$ to an integer value. The rounding operation can be done using a variety of operations (such as Floor, Ceil, Nearest-Neighbour, or Stochastic Rounding (Wang et al., 2018)), but will result in a loss of information either way, with a rounding error that decreases as we increase the parameter $b$: $\Delta \sim 2^{-b}$. If the value of $x$ is sufficiently small $|x| < 2^{-b}$, the quantization noise will exceed the represented value ($\Delta > |x|$) and we have no way to accurately represent the value of $x$. We will refer to this event as *underflow*. Second, we have a limited range for representation, that increases with the number of bits $B$ and decreases with $b$. We refer to the event when $x$ is outside the range $(R_{\min}, R_{\max})$ as *overflow*, noting that the quantization error in this case is unbounded.

Integer quantization is a specific case of fixed-point quantization, where the exponent bias $b$ is set to 0. While we defined the exponent-bias $b$ to be an integer, it is important to note that non-integer values could have worked mathematically just as well to define valid quantization operations. The main benefit of choosing $b$ to be an integer is the efficiency of computing power-of-two multiplications in hardware.

The main advantage of fixed point quantization comes from its relative simplicity. Integer multiplication (and addition) are generally considered to be cheaper on hardware when compared with floating point operations.

## 2.3 Floating point Quantization

Given a full-precision scalar value $x$, number of mantissa bits $M$, number of exponent bits $E$, and an integer $b$ (exponent-bias), we define the floating point (Dekker, 1971) quantization $M/E$:

$$
\begin{aligned}
s &\equiv \tfrac{1}{2}\left(1 - \operatorname{sign}(x)\right), \quad e \equiv \lfloor \log_2(|x|) \rfloor \\
m &= 2^{-M}\operatorname{Round}\left(2^M\left(|x|2^{-e} - 1\right)\right) \\
R_{\mathrm{OF}} &\equiv 2^{2^E - b - 1}\left(2 - 2^{-M}\right), \quad R_{\mathrm{UF}} = 2^{-b}
\end{aligned}
\qquad
Q^{\mathrm{FLOAT}}_{M,E,b}(x) \equiv (-1)^s
\begin{cases}
R_{\mathrm{OF}} & |x| \geq R_{\mathrm{OF}} \\
0 & |x| < R_{\mathrm{UF}} \\
2^e\,(m+1) & \text{else}
\end{cases}
\tag{2}
$$

Note that $1 \leq |x|2^{-e} < 2$, due to the definition of $e$, which helps make sense of the quantization operation in Eq. (2). The total number of bits used for this representation is $B = M + E + 1$: 1 sign bit ($s$), $M$ mantissa bits ($m$) and $E$ exponent bits ($e$). As we can see, floating point representation can cover a larger range of values when compared with a fixed point representation that uses the same amount of bits and exponent bias, ($R_{\mathrm{OF/UF}}$ depends on $2^{2^{\pm E}}$ while $R_{\max}, R_{\min}$ depends on $2^B$), reducing the occurrence of overflow and underflow events.

Unlike fixed-point representation, which had a fixed bound for quantization error ($\Delta \sim 2^{-b}$) within the represented range, the quantization error for floating point representation varies, depending on the magnitude of $x$: $\Delta \sim 2^{e-M}$. As a direct result, floating point's arithmetic also adds additional complexity, in the form of *swamping* Higham (1993). When performing an addition over two floating points values $\bar{z} = z_1 +_{\text{(FP)}} z_2 \equiv Q^{\mathrm{FLOAT}}_{M,E,b}(z_1 + z_2)$, it is possible that the precision of $\bar{z}$ will not be sufficient for full-representation of its summands, causing the least significant bits to be swamped out — resulting in a 'noisy' addition operation. In the extreme case, denoted as *Full-Swamping*, if $|z_1| > 2^{M+1}|z_2|$, $z_2$ is swamped out entirely, so $\bar{z} = z_1$ despite $z_2$ being non-zero. In contrast, fixed-point addition will always be exact, as long as the sum remains within the representation range (no overflow).

## 2.4 Low Bit-Width Accumulators

When performing a general matrix multiplication (GEMM) operation, (e.g. matrix-multiplication, or convolution), each individual scalar computed during the operation can be expressed as the sum of product pairs

$$
y = \sum_{i=0}^{N-1} x_i w_i.
\tag{3}
$$

Here, $y$ is a scalar component of the output tensor of the GEMM operation, $N$ is the accumulations size (i.e., the number of summands used per scalar output), while $\{x_i\}_{i=0}^{N-1}$ and $\{w_i\}_{i=0}^{N-1}$ are two series of scalar inputs used for the calculation of $y$. The values in both series originate from the input tensors, but the exact mapping, from tensors to series, will depend on the performed operation (see Appendix A for more details). Due to the common structure of the multiply-accumulate operation, hardware implementations of GEMM operation often rely on the fundamental Fused Multiply-Add (FMA) operation, defined as $\mathrm{FMA}(x, w, s) \equiv x \cdot w + s$, with $x, w, s$ being scalars. Our goal in this work will be to decrease the cost of the FMA component.

Previous discussed methods, such as W/A quantization, have been helpful in reducing the cost of the multiplication of FMA. In contrast, the accumulation component of FMA has been studied to a much lesser extent. In (Wang et al., 2018), the authors show that training a neural network with FP16 accumulators can result in noisy training, with a modest loss of accuracy. To mitigate this, the paper recommends chunk-based accumulation and floating-point stochastic rounding. Chunk-based accumulation changes the order of accumulation, while stochastic rounding is a method where a small, random noise is added to the result of high-precision summation, before the result is cast to a low-precision representation. While successful at closing the gap (e.g., for ResNet18 on ImageNet), both methods may prove difficult to implement on modern hardware. Specifically, the order of accumulation on DL accelerators will usually depend on their block architecture and is not easily configured. Moreover, stochastic rounding requires an implicit addition operation, which is projected to increase the cost of hardware addition, negating the benefit of using LBAs.

Sakr et al. (2019) examined the effect of low precision accumulators on training through the *accumulation variance* statistic, which they theoretically derive, given several statistical assumptions on the distribution of the summands. In Ni et al. (2020), the authors propose WrapNet, where the

additions are performed with 8 and 12 integer accumulators with wrap-around. WrapNet is shown to perform complex inference tasks (e.g. ImageNet classification) with extreme quantization (e.g., 7 bits activations, 2 bit weights, and 12 bits accumulators), but it does suffer a noticeable accuracy degradation in this setup, for tasks such as ImageNet classification.

Although mostly experimental, FP16 accumulation was integrated in the design of several commercial products (Agrawal et al., 2021), including the tensor cores in the Hopper architecture (NVIDIA) Andersch et al. (2022).

## 3 FINE-TUNING NEURAL NETWORKS WITH LOW-BIT ACCUMULATORS

One key difference between W/A quantization and quantization of the accumulators is that accumulation is an internal FMA operation, which is not generally visible to the software user. To simulate the effect of quantized FMA component, we implement the GEMM operations (convolution/ matrix multiply) in CUDA, where the FMA operation is replaced with our custom FMAq operation:

$$\text{FMAq}(x, w, s) \equiv Q_{\text{acc}}\left(Q_{\text{prod}}\left(x \cdot w\right) + s\right), \tag{4}$$

as illustrated in Fig. 1. In all experiments, we use a constant chunk size of 16, based on the sizes exposed to the user of NVIDIA's tensor cores. It is important to highlight that the product and

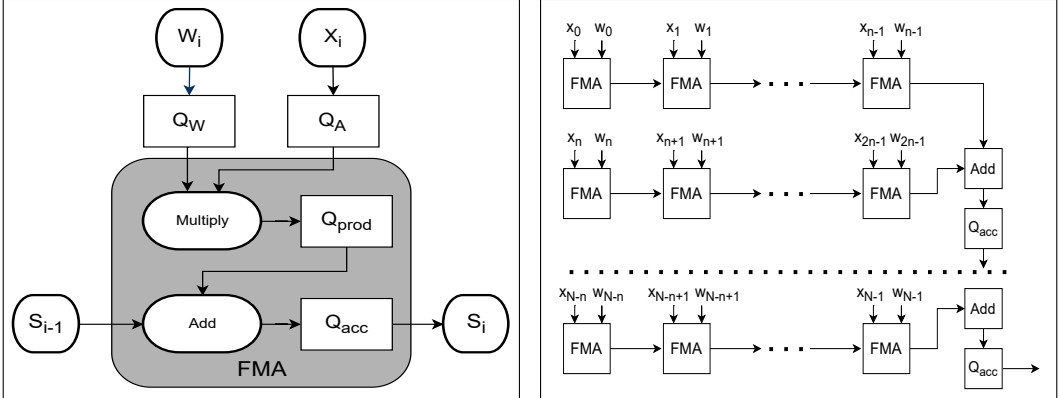

Figure 1: Left: an illustration of quantized FMA component, as simulated in our work. Unlike the W/A quantization operations ($Q_W(w), Q_A(x)$) that can be efficiently performed in software, $Q_{\text{prod}}$ and $Q_{\text{acc}}$ are explicitly internal hardware operations, intended to simulate the logic of a cheaper hardware component. Right: Illustration of chunk-based accumulation, with chunk base of $n$. Chunk-based accumulation is useful for reducing error caused by swamping, but the chunk size is not easily configured and will usually depend on the architecture design of the systolic array.

accumulator quantization functions ($Q_{\text{prod}}$ and $Q_{\text{acc}}$) are intended to simulate the hardware, rather than suggest an implementation for it. Breaking down the FMA to components in hardware would, in practice, undermine its efficiency — as it will no longer be 'fused'. Taking this into account, $Q_{\text{prod}}$ and $Q_{\text{acc}}$ must remain simple and computationally efficient. For example, 'round to nearest' and stochastic rounding methods, which are taken for granted for W/A quantization, will not be available to us during inference, as their hardware implementation would still perform addition internally with a higher number of bits. Our quantization will instead rely on the simple 'floor' operation, implemented in software via bit-mask. For Hardware analysis of our implementation, see Appendix E.

As discussed in section 2.3, floating point quantization can be broken down into 3-distinct events: *underflow*, *overflow* and *swamping*. Eventually, our low-precision model will have to handle all three events. We will, however, start by examining their individual properties, as displayed in Tab. 1.

Our main insight from Tab. 1, is that underflow events are expected to have the least significant effect over the network output (They have the lowest absolute error, since the default value for the exponent bias $b$, as used by the common FP32/FP16 definitions, is $b = 2^{E-1}$.). In Fig. 2, we evaluate the correctness of this claim, and show that the wide-scope loss-landscape of an LBA ResNet is barely affected when we ignore UF events. And yet, the large relative error induced during underflow

| Event | Condition | Key Parameters | Absolute Error (bound): $\Delta = |Q(x) - x|$ | Relative Error: $\frac{\Delta}{|x|}$ |
|---|---|---|---|---|
| Overflow (OF) | $|x| \gtrsim 2^{2^{E-b}}$ | $E, -b$ | $\infty$ | $(0\%, \infty)$ |
| Underflow (UF) | $|x| < 2^{-b}$ | $E, b$ | $2^{-b}$ | $100\%$ |
| Swamping | No OF/UF | $M$ | $2^{\lfloor \log_2(|x|) \rfloor - M}$ | $\left[ 2^{-M-1}, 2^{-M} \right]$ |

Table 1: Properties of each type of floating-point quantization event.

(small elements are effectively replaced with zero), will cause significant optimization errors for gradient-based methods: During fine-tuning, we can expect the magnitude of the weight updates to be proportional to the magnitude of the corresponding weights, causing the underflow region to be particular hard region to 'escape' from. Values that are stuck at underflow are effectively excluded from the training since the forced value of zero prevents them from learning meaningful correlations. (see Appendix F for more details.)

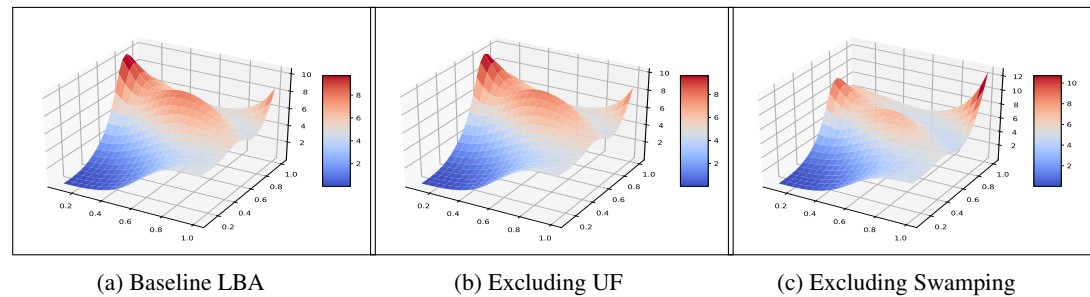

(a) Baseline LBA          (b) Excluding UF          (c) Excluding Swamping

Figure 2: Wide scope loss landscapes Li et al. (2018) of an LBA resnet50, using pre-trained ResNet50 weights (CIFAR10, FP32). Here, we compare the qualitative effect of different components in floating points quantization over the network output: In (a), we use a complete implementation of FP quantization during convolution accumulation, with 7 Mantissa and 4 Exponent bits. In (b), we repeat the previous experiment but ignore underflow events during quantization. For comparison, in (c), we repeat the original experiment, but add 16 additional bits to the mantissa, greatly diminishing the effect of swamping, without affecting the role of underflow. All landscapes appear similar, but while the effect of excluding swamping events (c) is visible, the loss landscapes of networks with (a) and without (b) underflow are hardly distinguishable.

Therefore, we propose the following method: Starting off with the weights of a pre-trained network (trained in full-precision), we will design a network that utilizes quantized FMA for forward-propagation, excluding underflow events, and perform a standard gradient-based optimization (i.e. Stochastic gradient decent, while keeping the backward implementation of each operation as it was with full-precision FMAs). Once we converge to some accuracy value, we will enable the underflow implementation and proceed with further fine-tuning.

As seen in Tab. 1, the exponent bias ($b$) can be configured to control underflow and overflow events, with a clear trade-off between the former and the latter. Previous works Kuzmin et al. (2022) have made the insight, that the default value $b = 2^{E-1}$ is not always suitable for neural networks. For our purposes, we note that the different quantization functions $Q_{\text{prod}}$ and $Q_{\text{acc}}$ as seen in Fig. 1, are likely to require different ranges for representation: Assuming the product terms $u_i = w_i x_i$ are i.i.d, the accumulator's value will follow the central limit theorem, and is therefore more likely to reach overflow, resulting unbounded quantization noise. To try and avoid this scenario, our setup will give a smaller exponent bias to the accumulator. In our experiments, we use a relative factor based on the chunk-size, so that $b_{\text{acc}} = b_{\text{prod}} - \frac{1}{2} \log_2 (\text{Chunk-Size})$. Following the same reasoning, one may suggest that the exponent bias should depend on the sequence number in which the FMA is applied within every GEMM operation. Nevertheless, for the context of this work, we will treat all FMA units as homogeneous, with the same exponent bias.

## 3.1 Experiments: Image Classification

For our first set of experiments, we aim to check the effect low-bit accumulators have on residual neural networks He et al. (2016). For each experiment, we use the standard ResNet architecture and replace each GEMM operation used during forward-propagation (convolution and matrix multiplication) with our custom implementation, as described in section 3. With no adjustments, the effect of these changes on the network accuracy (zero-shot), can be severe, as we show in Appendix B.

For $Q_{\text{prod}}, Q_{\text{acc}}$, we used the same amount of mantissa and exponent bits, $M = 7, E = 4$, a setup we will denote as $M7E4$. For overflow, we used the exponent biases: $b_{\text{acc}} = 10, b_{\text{prod}} = 12$, but disabled underflow events for the first part of the experiment. After loading the networks with pre-trained weights, we proceed to train the network for 5 epochs, using Adam optimizer with a learning rate of $\eta_0 = 10^{-6}$, and a cosine scheduler, so that $\eta_5 = 10^{-8}$). Then, we enable underflow events and run a fine-tuning again for a single epoch, using a reduced learning rate of $\eta_{\text{UF}} = 10^{-7}$. To evaluate the benefit of the two-staged fine-tuning, we also ran the same experiment with a single stage, where underflow is enabled for 10 epochs. The baseline numbers were obtained by repeating the fine-tuning process in a non-LBA setup, which resulted in an improvement of up to $0.65\%$ over the zero-shot accuracy. Our full setup and implementation are detailed in Appendix C. The results of this experiment are presented in Tab. 2.

| Model | Baseline | 1-stage | no UF* | no UF $\rightarrow$ with UF |
|---|---|---|---|---|
| ResNet18 | 70.23% | 69.94% | 70.01% | 70.06% |
| ResNet34 | 73.87% | 73.64% | 73.61% | 73.45% |
| ResNet50 | 76.80% | 74.70% | 76.60% | 76.40% |

Table 2: Top-1 Accuracy results: Fine-tuning ResNets with low-bit accumulators for ImageNet classification.
*Intermediate Stage: Both training and evaluation are done without underflow.

For LBA ResNets with full-precision W/A, our results indicate that the models we suggest can train surprisingly well even without a dedicated fine-tuning regime. The dual-stage approach (Training without UF first and enabling it later) only shows clear benefit, so far, in the case of the larger, ResNet50 model. That being said, scaling the method for larger models is important, and tasks will only become more difficult from now on.

In order for a model with low-bit accumulators to be commercially viable, it is vital to show that quantized accumulation still works when the weights and activations are quantized. Therefore, our next set of experiments will test the feasibility of LBA ResNets in this setting. For weights and activations, we will use 8-bit floating point representation (Wang et al., 2018). Following the results presented in Kuzmin et al. (2022), we use $M4E3$ representation with flex-bias for both weights and activations, implemented using the *qtorch* library Zhang et al. (2019). For our flex-bias implementation, we evaluate the maximal exponent for each tensor during forward propagation, and use the maximal integer exponent bias that is sufficient to prevent overflows (single value per tensor). The results of fine-tuning LBA ResNets in this setup can be seen in Tab. 3, as well as a comparison of our results with previous works that also used lower-bit accumulators.

We note that a direct comparison between the methods based on final accuracy alone will not be valid: the method presented in Wang et al. (2018) is intended for quantized training, and includes several more quantized components, as well as several methods that are projected to reduce hardware efficiency. Meanwhile, Ni et al. (2020) proposes the cheapest implementation (Fewer bits for Weights and activations, Integer quantization), sacrificing model accuracy for hardware efficiency. Nevertheless, when aiming for cheaper inference, our LBA models were the only models to achieve accuracy on par with non-LBA models, while providing a cheaper alternative compared to models with standard accumulation.

## 3.2 Experiments: Language Models

To assess the capability of LBA language models, our next set of experiments will focus on the common Bert (Devlin et al., 2018) architecture, and the SQUAD (Question-Answering) task. In

| Model | Data Type | Weights | Activations | Accumulator | Top-1 Accuracy |
|---|---|---|---|---|---|
| **ResNet18** | | | | | |
| Baseline | FP | 32 | 32 | 32 | 70.23% |
| Baseline (FP8) | FP | 8 | 8 | 32 | 69.90% |
| Wang et al. (2018) | FP | 8 | 8 | 16 | 66.95% |
| Ni et al. (2020) | INT | 7 | 2 | 12 | 63.84% |
| Ours (1-stage) | FP | 8 | 8 | 12 | 69.54% |
| Ours (dual-stage) | FP | 8 | 8 | 12 | 69.70% |
| **ResNet34** | | | | | |
| Baseline | FP | 32 | 32 | 32 | 73.87% |
| Baseline (FP8) | FP | 8 | 8 | 32 | 73.49% |
| Ours (1-stage) | FP | 8 | 8 | 12 | 73.18% |
| Ours (dual-stage) | FP | 8 | 8 | 12 | 73.42% |
| **ResNet50** | | | | | |
| Baseline | FP | 32 | 32 | 32 | 76.80% |
| Baseline (FP8) | FP | 8 | 8 | 32 | 76.25% |
| Wang et al. (2018) | FP | 8 | 8 | 16 | 71.72% |
| Ours (1-stage) | FP | 8 | 8 | 12 | 74.15% |
| Ours (dual-stage) | FP | 8 | 8 | 12 | 76.22% |

Table 3: Top-1 Accuracy results: Fine-tuning ResNets with low-bit accumulators and FP8 weights and activations for ImageNet classification. Results are compared with similar models utilizing LBAs in the literature.

this case, fine-tuning a pre-trained model is already the standard. In contrast to our experience with residual networks, breaking down the fine-tuning process into separate phases was not, in general, beneficial for the accuracy of the larger models. The exponent biases we used for the different LBA models also had to be changed, to avoid overflow events. In table4, we compare the results of fine-tuning LBA Bert models with the results of fine-tuning non-LBA models, as described in C.2. While LBA Bert-small has a small ($\Delta_{f1} = 0.37\%$) performance degradation compared with the non-LBA model, the gap is closed completely for the Bert ($\Delta_{f1} = -0.09\%$) and Bert-Large ($\Delta_{f1} = -0.26\%$).

| | Baseline | | LBA ($M7E4$) $b_{acc},b_{prod}$=7,9 | | LBA ($M7E4$) $b_{acc},b_{prod}$=8,10 | |
|---|---|---|---|---|---|---|
| Model | Exact (%) | f1 (%) | Exact (%) | f1 (%) | Exact (%) | f1 (%) |
| Bert-Small | 71.32 | 80.96 | 70.88 | 80.24 | 71.35 | 80.59 |
| Bert-Base | 79.84 | 87.53 | 79.60 | 87.62 | 79.80 | 87.52 |
| Bert-Large | 83.22 | 90.40 | 82.97 | 89.97 | 83.25 | 90.66 |

Table 4: SQUAD v1 fine-tuning for LBA-Bert models.

Inspired by our LBA-Bert model results (which were favorable toward larger models), we tested our LBA-aware fine-tuning method on the LLama-v2-7B model (Touvron et al., 2023). We used the same settings and scripts as QLoRA paper (Dettmers et al., 2023), which uses frozen 4-bit weights with an additional trainable low-rank matrix in BF16. To measure performance on a range of language understanding tasks, we used the MMLU (Massively Multitask Language Understanding) benchmark Hendrycks et al. (2020), a multiple-choice benchmark covering 57 tasks. The fine-tuning was done over the Open Assistant (OASSA1) dataset Köpf et al. (2023) using official training scripts found in the QLoRA code (i.e., llama2_guanaco_7b). We report 5-shot test accuracy in tabel 5.

| Model | Baseline | $M10E5$ | $M6E5$ | $M7E4$* |
|---|---|---|---|---|
| LLamma v2 (OASSA1) | 45.3 | 45.4 | 44.3 | 45.1 |

Table 5: MMLU 5-shot test accuracy with and without LBA, for QLORA+ LLama v2 (7B parameters). * For runs with 4 exponent bits, we used dynamic (per-layer) exponent-bias.

## 4 BELOW 12 BITS: FINE-GRAINED GRADIENT FOR LOW BIT ACCUMULATORS

Thus far, we have shown that 12 bits are sufficient for inference in a variety of deep neural networks. However, the simple methods described in section 3 are not sufficient for training neural networks with lower amounts of accumulation bits.

For example, a shallow fully-connected DNN trained over MNIST, will fail when using a $M4E3$ accumulator, even when excluding underflow events. The cause of the failure is known as it is similar to quantization failures in other areas of deep neural network: Quantization changes the output of the network during forward pass, and when the change is significant enough, it is no longer feasible to rely on the gradients of non-quantized operations for optimization. Of course, we cannot use the "real" gradients with respect to quantized operation, since they are zero almost everywhere.

The common solution to this problem, with relation to the quantization of weights and activations, is to replace the derivative of the quantized function with a Straight-Through-Estimator (STE). In our case, we would like to use the STE with respect to the derivatives of the quantizers $Q_{\text{acc}}$ and $Q_{\text{prod}}$ inside the FMAq operation from Eq. (4). So far in this work, we used the naive "identity STE" (Bengio et al., 2013) which makes the replacement " $\frac{d}{dx}Q(x)$ " $= 1$ (we will use the quotation marks to denote an STE replacement of a derivative). However, the more common STE for quantization zeros out gradients outside of the representation range (Hubara et al., 2017). For the quantizers in Eq. (1) and Eq. (2), we get:

$$ \text{``} \frac{d}{dx}Q_{B,b}^{\text{FIXED}}(x) \text{''} = \mathbf{1}(R_{\min} < x < R_{\max}) \quad ; \quad \text{``} \frac{d}{dx}Q_{M,E,b}^{\text{FLOAT}}(x) \text{''} = \mathbf{1}(|x| < R_{\text{OF}}), \qquad (5) $$

where we defined $\mathbf{1}(\cdot)$ as the indicator function which is equal 1 if its input is 'true' and zero otherwise. Many alternative forms of STEs exist and have been studied in the context of W/A quantization. The implementation of STEs for LBA networks, on the other hand, has several additional difficulties.

The first, most immediate problem, is that the values of the inputs of the quantization functions within the FMAq ($Q_{\text{acc}}$ and $Q_{\text{prod}}$) are not exposed to the software or stored in memory during forward propagation. Saving these internal values is generally not feasible, since the quantization operation occurs in each FMAq, and the number of FMAqs in DNNs typically exceeds the size of weights and activations by many orders of magnitude. However, if the hardware operation is deterministic and well-known, we found we are still able to use software for re-computation of the GEMM operation, to retrieve the required values during backpropagation (1 bit per operation). Such a re-computation operation is expensive (training time is doubled, at the very least), and so far feasible only in fully connected and attention layers (not convolutional layers). To the best our knowledge, this is the first time backpropagation is used on the full computational graph of the summation operation.

Another possible problem for using standard STEs for the accumulation process stems from the recursive nature of the summation operation. The STE in equation Eq. (5) sets the corresponding gradient of any overflowing value to zero. As explained in Appendix D, if this STE is used for the accumulator's quantization function, each overflow event will eliminate the gradients of all previously accumulated product pairs. Lastly, another possible problem is that, for floating point summation, other events besides overflow can potentially be important when estimating the gradient.

Motivated by the last two potential problems, in appendix D, we propose, describe, and justify the practicality of several alternative methods for estimating the gradients of $\text{FMAq}(x, w, s)$. The different methods use different types of STE: *OF* passes zero on overflow of $Q_{\text{acc}}$ (using Eq. (5), while *DIFF* passes zero on overflow, underflow, and full-swamping events of the FMAq. We also distinguish between a method where we apply identity STE with respect to the partial sum $s$, and the non-identity STE over the product-pair $(x, w)$ (a.k.a *Immediate*), to the standard method, where the STE is applied with respect to all inputs $(x, w, s)$ (a.k.a *Recursive*). For example, defining $z \equiv \text{FMAq}(x, w, s) = Q_{\text{acc}}\left(Q_{\text{prod}}\left(x \cdot w\right) + s\right)$ and $\epsilon_1, \epsilon_2$ as some small constants, we get:

$$ \text{Immediate / DIFF:} \quad \text{``} \frac{dz}{ds} \text{''} = 1 \quad ; \quad \frac{1}{w} \text{``} \frac{dz}{dx} \text{''} = \frac{1}{x} \text{``} \frac{dz}{dw} \text{''} = \mathbf{1}\left(\frac{|z - s|}{|xw| + \epsilon_1} > \epsilon_2\right), \qquad (6) $$

$$ \text{Recursive / OF:} \quad \text{``} \frac{dz}{ds} \text{''} = \frac{1}{w} \text{``} \frac{dz}{dx} \text{''} = \frac{1}{x} \text{``} \frac{dz}{dw} \text{''} = \mathbf{1}(|Q_{\text{prod}}\left(xw\right) + s| < R_{\text{OF}}) \qquad (7) $$

In Tab. 6, we compare the accuracy achieved using the proposed STE variants over the MNIST dataset. We see that such fine-grained gradient methods can indeed enable high accuracy in models with only 8-bit accumulators.

| STE | Underflow | Accuracy (Top-1, %) | STE | Underflow | Accuracy (Top-1, %) |
|---|---|---|---|---|---|
| Baseline | - | 98.65 | Immediate / OF | Yes | 98.47 |
| Identity | Yes | 18.28 | Immediate / DIFF | Yes | 11.35 |
| Identity | No | 18.28 | Immediate / DIFF | No | 97.67 |
| +Identity* | Yes | 42.28 | Recursive / OF | Yes | 98.47 |

Table 6: Training a fully-connected NN with 8-bit ($M4E3$) accumulators for MNIST classification. The reported accuracy matches the final accuracy of the experiment. The model's loss does not converge when using naive (Identity) STE for accumulation. Full details in Appendix C.3.
*The mantissa for the accumulator was extended by 2 additional bits in this run.

As we saw in the case of residual neural networks (Tab. 2 and 3) with 1-stage training, successful implementation of LBA is not guaranteed to scale to larger models. To evaluate the quality of our estimated gradients, we would like to compare the optimization of the different approaches. To that end, we train a small LBA transformer from scratch for masked language modeling, over a modest-sized dataset ($200K$ rows), for 15 epochs. In Tab. 7, we compare different STE variants for a variety of very-low precision accumulators.

| Accumulator | Identity (%) | Recursive / OF (%) | Immediate / OF (%) | Immediate / DIFF (%) |
|---|---|---|---|---|
| FP32 | 51.31 | - | - | - |
| $M3E3$ | 20.86 | 19.20 | 14.80 | **24.60** |
| $M4E3$ | 13.88 | 39.57 | 37.23 | **41.94** |
| $M5E3$ | 9.47 | 45.28 | 44.76 | **50.12** |
| $M6E3$ | 14.71 | 46.17 | 46.13 | **50.03** |
| $M3E4$ | 15.2 | 15.15 | 15.43 | **25.53** |
| $M4E4$ | **42.93** | 42.81 | 42.81 | 41.50 |
| $M5E4$ | 47.87 | **48.76** | **48.76** | 47.93 |

Table 7: Accuracy of LBA transformer for the task of Masked Language Modelling ($200K$ rows), when using different STEs for the accumulator operation. Full details of the experiments are available in Appendix C.4.

Based on our results for training masked language models, using fine-grained STEs becomes crucial when the number of accumulation bits is decreased below $M = 4$ or $E = 4$ (hence, this includes all possible FP8 formats). While successful at improving the optimization, none of the STEs we have tried were successful at closing the gap with the baseline completely, when extreme accumulator quantization was applied. Out of the three proposed STEs, we recommend Immediate/ DIFF STE, which generally achieved better accuracy in the areas where naive, identity STE was insufficient, despite its higher cost. The Immediate/ DIFF STE may also prove more suitable in cases where the exact behavior of the FMAq is unknown (i.e., 'black-box') since its definition is agnostic to the FMAq internals.

## 5  DISCUSSION

The quantization of the accumulator in deep neural networks is a hard but necessary task in the effort to improve neural networks' efficiency, reduce cost, and cut down carbon footprint. Despite the many difficulties involving the training, the implementation, and the theoretical analysis of networks with low-bit-accumulators, our results show that LBA networks are surprisingly easy to fine-tune. By applying simple optimization methods over pre-trained networks, we show it is possible to adjust the models for inference with cheaper hardware, that utilizes 12 bits accumulators. When the accumulators bit width is further reduced we alleviate the accuracy degradation by using fine-grained approaches for estimating the gradient.

ACKNOWLEDGMENTS

The research of DS was Funded by the European Union (ERC, A-B-C-Deep, 101039436). Views and opinions expressed are however those of the author only and do not necessarily reflect those of the European Union or the European Research Council Executive Agency (ERCEA). Neither the European Union nor the granting authority can be held responsible for them. DS also acknowledges the support of the Schmidt Career Advancement Chair in AI.

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

## A    GENERAL MATRIX MULTIPLICATION: EXAMPLE

In section 2.4, we defined the FMA operation, and presented Eq. (3) as a general formula for all GEMM operation. It is worth taking a moment to illustrate the connection between the known tensor operations and the formula.

For example, let us look at the simple case of matrix-multiplication ($Y = XW^T, X \in \mathbb{R}^{d_0 \times d_1}, W \in \mathbb{R}^{d_2 \times d_1}$). Here, if we wish to calculate the scalar $y = Y_{kl}$, we can use the mapping: $x_i = X_{ki}, w_i = W_{li}$. In this case, all values of $X$ and $W$ were used exactly once in the calculation of $Y$. This is not always the case, however. In batch matrix multiplication, values of $W$ will be used multiple times, paired with values of $X$ of different batch dimension.

In convolution, the same values of $W$ will be used to calculate every scalar in the same output channel, and the neuron in the input channel may be used to calculate a multiple values in multiple output channel. This level of repetition will be, in part, what prevents us from using fine-grained STE methods on convolutional neural networks, in Sec. 4.

## B    EFFECT OF QUANTIZED FMA ON ZERO-SHOT ACCURACY

To give the reader a sense of the effect of low bit accumulators on deep neural networks, we include Tab. 8, where we measure the zero-shot accuracy of different ResNet architectures, pretrained with full-precision, after replacing all FMA components with FMAq (as described in C).

| Mantissa Effect | | | | | |
|---|---|---|---|---|---|
| Model | Baseline | M10E5 | M9E5 | M8E5 | M7E5 | M6E5 |
| ResNet18 | 69.75 | 69.50 | 68.95 | 66.70 | 57.09 | 20.49 |
| ResNet34 | 73.31 | 73.17 | 72.68 | 70.46 | 60.07 | 17.19 |
| ResNet50 | 76.12 | 75.95 | 75.57 | 73.70 | 64.94 | 19.48 |
| Exponent Bias Effect (M7E4) | | | | | |
| Model | $b = 8$ | $b = 9$ | $b = 10$ | $b = 11$ | $b = 12$ | $b_{\text{acc}}, b_{\text{prod}} = 10, 12$ |
| ResNet18 | 55.68 | 60.64 | 60.00 | 58.84 | 56.96 | **60.14** |
| ResNet34 | 50.80 | 63.30 | **63.88** | 62.46 | 59.90 | 63.65 |
| ResNet50 | 26.41 | 64.25 | **68.69** | 67.57 | 66.12 | 68.49 |

Table 8: Zeroshot Accuracies for LBA-ResNets, with weights of pre-trained, full precision ResNets [%]

The accuracies presented in Tab. 8 illustrates well why $M7E4$ quantization was chosen: Increasing the mantissa below $M = 7$ bits would result a much lower zero-shot accuracy, too far for proper fine-tuning. Likewise, reducing the number of bits to $E = 4$ already resulted lower accuracy due to overflow and underflow events, as indicated by the effect of the exponent bias. For example, the default exponent bias for $E = 4$ is $b = 8$, and using it for the accumulator in Resnet50 results in a significant degradation in accuracy. A small increase to $b = 9$, increases both underflow and overflow thresholds by a factor of 2 and is sufficient for increasing the accuracy by almost $40\%$.

## C    EXPERIMENTS IMPLEMENTATION DETAILS

### C.1    IMAGENET

Each of the ImageNet experiments were performed on a single server, containing 8 NVIDIA GPUs (RTX 2080 Ti, RTX A6000). We used a total mini-batch size of 256, equally divided across the 8 workers. For the training datasets, we used the standard *RandomResizedCrop* and *RandomHorizontalFlip* augmentations only. With no quantization, our model architecture was identical to the standard *torchvision* architecture, for all ImageNet models, while our custom GEMM kernels were used to override all forward GEMM operations (convolutions and matrix multiplications). For optimization, we used the Adam optimizer, with the hyperparameters $\beta = (0.9, 0.999), \epsilon = 10^{-8}, \lambda = 10^{-4}$. Dropout was not used. As mentioned in the main text, we used cosine scheduling, the parameters of which depend on the phase in which it was used. We used 10 epochs in the 1-stage compared

with 5 epochs for the dual-stage to support our claims that the gaps between the methods (where they exist) are not simply a result of better hyperparameters. The epoch count was initially chosen due to time-constraints, and was kept since the benefit of running more epochs was small.

For W/A quantization, we used the *qtorch* (Zhang et al., 2019) library, which provides reliable quantization functions. Weights quantization was applied during every optimization step, while the activations were quantized using dedicated modules, preceding all convolutions, except the first one, and the downsample convolutions. The input of the final fully-connected layer was not quantized as well, in accordance with prior works. The quantization function we applied used stochastic-rounding (which is not considered expensive in this case, as we are not implementing the FMAq internals and the number of quantized components is significantly lower). No other components (e.g. Gradients or Momentum) were quantized since our solution is only aimed at inference.

In addition to hyperparameters used in this experiment, we have also ran a set of experiments using fixed-learning rates (no cosine annealing). In the other set, we tested a few initial learning rate values (1E-7, 3E-8, 1E-8) for few epochs, and used enough epochs to reach convergence (for training accuracy/loss). The results in this regime were slightly better than the results published in the paper: For 8bit quantized ResNets with 4ME3, we achieved 69.6% for Resnet18, 73.48% for ResNet34 and 76.35% for ResNet50. However, this required more epochs and finer-tuned hyperparameters (different models used different learning rates). In the paper, we used the regime with cosine annealing since it was more robust to hyperparameter changes.

## C.2 SQUAD

For the SQUAD fine-tuning experiment, we use 8 NVIDIA GPUs (RTX 2080 Ti, RTX A6000). We used the SQUAD training script of the transformers library (Wolf et al., 2019), while using our custom, LBA-model. In our LBA model, all fully connected layers and matrix multiplication operations were modified to use LBA GEMM operations during forward propagation, with the exception of the final fully connected layer (*qa-outputs*). For pre-trained models, we used either *bert-base-uncased* for Bert or *prajjwal1/bert-small* for Bert-small. For optimization, the Adam optimizer, with 1000 warmup steps to a learning rate of $3 \cdot 10^{-5}$, from which we applied a cosine annealing scheduler. The batch size was configured to be 8. Our run was set for 20 epochs, but we applied early stopping once the model performance reached its peak (usually after $3 - 5$ epochs).

## C.3 MNIST

For each experiment with the MNIST setting, we used a single RTX 2080 Ti GPU with a mini-batch size of 16. Our neural network consisted of 4 fully connected layers (with LBA), and ReLU activations, with all hidden layers being 1024 neurons wide. Outside of the accumulator, all data types were with full precision. Dropout wasn't used (although it was shown to benefit the results slightly), and no data augmentation wasn't used during training. For optimization, we used Adam optimizer, with an initial learning rate of $10^{-3}$, with the hyper-parameters: $\beta = (0.9, 0.999), \epsilon = 10^{-8}, \lambda = 0.0$, and *StepLR* scheduler ($\gamma = 0.95$). We used 100 epochs per experiment, which was usually much more than needed for convergence or divergence.

To test the STE, we replaced the default linear operations with our custom implementation, this time also implementing a custom backward operation. During backpropagation, we used a new, cuda kernel that imitated the (deterministic) operation of the original GEMM operation (using the available weights and activations), but outputted a binary tensor, that indicated the value of all STEs involved in the operation (the type of STE was configurable). For recursive implementation, we modified the tensor ad-hoc to account for the recursive nature of the STE (although less efficient than the optimal implementation). After running the kernel, we used the output to adjust the computation of the weights/ neural gradients as described in section D. In this experiment, we used a fixed exponent bias of 5, which was shown to perform the best among all values in its vicinity.

## C.4 MASKED LANGUAGE MODELLING

Each of the Masked Language Modelling (MLM) experiments was performed on a single server, containing 8 NVIDIA GPUs (RTX 2080 Ti, RTX A6000, or A100). Our tests were run over the *oscar*: *unshuffled-original-af* dataset, with a single tokenizer we trained over the same dataset (vocabulary

size of $1000$). The dataset was chosen due to its moderate size ($200K$ rows), being difficult enough to show gaps in convergence while allowing us to perform meaningful optimizations with simulation kernels in moderate time. For the transformer, we used the Bert architecture, with the hidden size of $512$, $2$ hidden layers, $4$ attention heads, and maximum position embedding of $1024$ (All other parameters were according to *transformers* library defaults). We used the available Huggingface infrastructure (Wolf et al., 2019) to train/ evaluate the model, with Adam optimizer, an initial learning rate for $10^{-3}$, a drop-on-plateau scheduler (evaluating every $250$ step, $\gamma = 0.1$), and a global mini-batch size of $64$. In practice, the drop-on-plateau scheduling was only applied to 'failed' runs, to give them another shot for optimization, with no success (They did not converge, even well passed the $15$ specified epochs). When the number of exponent bits was set to $E = 3$, we used a fixed exponent bias of $b = 6$ for the product and accumulator.

## D  GRADIENT ESTIMATION FOR LBAS

Following the general equation for GEMM operation (Eq. (3)), the operation can be expressed, using the recursive expression:

$$S_0 = 0; \quad S_{i+1} = \text{FMA}\left(x_i, w_i, S_i\right); \quad y = S_{N-1} \tag{8}$$

In this example, we add the values of the product to the intermediate accumulator sum, ($S$), in a sequential manner. Different orderings of the FMA operation are possible and can have an effect on the output (i.e., floating point addition is not commutative as 'ideal' addition, due to *swamping*).

Let us write the recursive expression in Eq. (8) explicitly

$$S_i^q \equiv \text{FMAq}\left(x_{i-1}, w_{i-1}, \text{FMAq}\left(x_{i-2}, w_{i-2}, \text{FMAq}\left(...\text{FMAq}(x_0, w_0, 0)\right)\right)\right); \quad y^q = S_{N-1}^q. \tag{9}$$

Our goal in this section is to find a good estimate for the derivative for $\frac{\partial y^q}{\partial x_i}$ and $\frac{\partial y^q}{\partial w_i}$.

### D.1  RECURSIVE STES

The first, and most obvious method to estimate the derivative is by using the common STE (Eq. (5)). Per our definition, FMAq contains two quantization functions. Our main concern, however, is for the post-accumulator quantization, $Q_{\text{acc}}$, and our first attempt will be to quantize it directly using a general STE function (STE : $\mathbb{R}^3 \to \mathbb{R}$). By doing so, we get the gradients:

$$\text{``}\frac{dy^q}{dS_i^q}\text{''} = \frac{1}{w_i}\text{``}\frac{dy^q}{dx_i}\text{''} = \frac{1}{x_i}\text{``}\frac{dy^q}{dw_i}\text{''} = \frac{dy^q}{dS_{i+1}^q}\text{STE}\left(x_i, w_i, S_i^q\right), \tag{10}$$

which, when expanded upon, will give us:

$$\text{``}\frac{dy^q}{dS_i^q}\text{''} = \prod_{k=i+1}^{N} \text{STE}\left(x_k, w_k, S_k^q\right). \tag{11}$$

Eq. (11) reveals an additional, possible issue for using standard STEs for the accumulation process. Usually, when applied over an overflowed activation neuron, the STE in equation Eq. (5) will set the corresponding neural gradient to zero. If the same STE is used for the accumulator's quantization function, as per Eq. (11), each overflow event will eliminate the neural gradients of all previously accumulated product-pairs. Still, this approach may help us calculate a more accurate gradient, provided that conditions in which the STE returns $0$ are not commonly met. We will denote the approach in Eq. (11) as *Recursive*.

To perform the recursive correction to the gradient, all we really need to know is the last index (if any), in which the accumulator was overflown. While this may be more complex in cases where the values are added in non-sequential ordering, this is still a feasible calculation to perform, with modest-sized output. Computationally, calculating the underflow indexes is no more difficult as a task than computing the original GEMM operation, and this calculation (as well the calculation that will be presented in the following section), can be done during backpropagation, to avoid using essential memory for a long term.

## D.2 IMMEDIATE STEs

To avoid setting too many gradients to zero, we suggest an alternative approach. First, we will re-write Eq. (9) as:

$$S_i^q = \alpha_0 x_0 w_0 + \alpha_1 x_1 w_1 + \alpha_2 x_{2j} w_2 + ... + \alpha_{i-1} x_{i-1} w_{i-1} \,, \tag{12}$$

where

$$\alpha_i \equiv \frac{\text{FMAq}\,(x_i, w_i, S_i^q) - S_i^q}{x_i w_i}. \tag{13}$$

If $x_i = 0$ or $w_i = 0$, we will define $\alpha_i = 0$ for simplicity. Recall $S_i^q$ here is the value of the "quantized" accumulator in step $i$ (Eq. (9)). From its definition, $\alpha_i$ is the correction we make to the product $x_i \cdot w_i$, to account for the FMA quantization error. Our choice to express the correction this way is based on the assumption that for most steps, $|S_i^q| \gg |x_i \cdot w_i|$. This is true, because $S_i^q$ is, approximately, the accumulated sum of many such products. During a floating point addition, the bits of the lesser value will be the first to be swamped out, and thus we entangle the quantization error with this component.

Moving on to the gradients, we are interested in the gradients of the operation inputs, $\frac{dy^q}{dx_i}$ and $\frac{dy^q}{dw_i}$. We can use the chain rule to get:

$$\frac{dy^q}{dx_i} = w_i \alpha_i + \sum_{k=i+1}^{N-1} \frac{d\alpha_k}{dx_i} x_k w_k. \tag{14}$$

The exact expression we got for the gradient remains complex, since $\frac{d\alpha_k}{dx_i} \neq 0$, for $k > i$. Nevertheless, moving forward, we will take the approximation that $\forall k, \frac{d\alpha_k}{dx_i} = 0$. i.e., we neglect the cumulative effect that any individual scalar value has on the quantization correction we make in the following steps of the GEMM operation. We then get:

$$\frac{dy^q}{dx_i} = w_i \alpha_i, \qquad \frac{dy^q}{dw_i} = x_i \alpha_i \tag{15}$$

Eq. (15) suggests a correction we can make to the standard backpropagation operation, which we will denote as an *Immediate* STE. However, to make any use of this correction, we must first have the values $\alpha_i$. In terms of computation, calculating $\alpha$ is quite similar to performing the original GEMM operation. In terms of memory, however, $\alpha_i$ scales with the number of overall FMA operations. This is feasible in the case of fully connected operations, but not for GEMM operations that include a large amount of shared weights. To make sure the evaluation of $\alpha_i$ by itself does not overburden the memory, it is possible to quantize the values of $\alpha_i$. By doing so, we get the equation:

$$\frac{1}{w_i} \text{``}\frac{dy^q}{dx_i}\text{''} = \frac{1}{x_i} \text{``}\frac{dy^q}{dw_i}\text{''} = Q_{B,0}^{\text{FIXED}} \left( \frac{\text{FMAq}\,(x_i, w_i, S_i^q) - S_i^q}{x_i w_i + \epsilon_1} \right). \tag{16}$$

where $\epsilon_1$ is a small constant, added with flexible sign to prevent invalid denominator. In our experiments, we have observed that the quantization of $\alpha_i$ does not harm the quality of the optimization process, and proceeded to binarize the value. The result is that we ended up suggesting an alternative STE to the one presented in Eq. (5), which is designed to address overflow only. We denote the new STE as *DIFF*:

$$\text{STE}^{\text{DIFF}}\,(x_i, w_i, S_i^q) = \begin{cases} 1 & \frac{|\text{FMAq}(x_i, w_i, S_i^q) - S_i^q|}{|x_i w_i| + \epsilon_1} > \epsilon_2 \\ 0 & \text{Else} \end{cases}$$

$$\text{STE}^{\text{OF}}\,(x_i, w_i, S_i^q) = \begin{cases} 1 & |Q_{\text{prod}}(x_i w_i) + S_i^q| < R_{\text{OF}} \\ 0 & \text{Else} \end{cases} \tag{17}$$

The DIFF STE is similar to the common Overflow STE, but is tuned to detect cases of full-swamping and cases of product underflow in addition to cases of overflow. In our experiments, we tested the *immediate* approach with both STEs. One unique advantage of the DIFF STE, is that is agnostic to the specific implementation of the FMAq component. Therefore, the DIFF STE remains relevant in

the general cases where the FMAq operation has an unspecified, black-box behavior, as common for hardware modules.

Our derivation in this section was done in respect to the sequential ordering of FMAq operations, which is not commonplace in hardware accelerators that try to achieve large degree of parallelism. A more typical case, presumably common for systolic-array-like accelerators, is the chunk-based accumulation, where the accumulation is performed in two hierarchies, as seen in Fig. 1. In our experiments, all simulated GEMM operations and gradient estimation used a chunk size of 16, which means that an operation with an accumulation width of $N$ is initiated with $\frac{N}{16}$ parallel operations (i.e., the first hierarchy), before aggregating the results (i.e., the second hierarchy). For example, in the case of recursive STE, every detection of OF or DIFF during re-computation will result in a '0' in all preceding operations, just as we saw for sequential accumulation. The only difference for parallel accumulation is that the hierarchy tree can expand in two directions (Like 1 (right), with all the arrows reversed).

## E  Hardware Analysis

In this section, we try to give an estimate for the effect of incorporation of LBA models on hardware cost (area/ power), by estimating the number of gates needed to implement qLBA with different levels of quantization.

Following an existing design of FMAq component (van Baalen et al. (2023),figure 2b), we adjusted the design for the case of FMAq with m/e quantization of weights and activations and M/E quantization of intermediate values (product, accumulator), and suggested the following gate counts, as seen in table 9. The gate counts are all based on the gate count assumptions listed in [van Baalen et al. (2023), appendix B], and common block designs.

| FMA Components breakdown | Gate Count |
|---|---|
| Exponent Adder | $(e - 1) \cdot C_{\text{FA}} + C_{\text{HA}}$ |
| Exponent Differ | $(\min(E, e + 1) - 1) \cdot C_{\text{FA}} + C_{\text{HA}} \cdot (1 + |e + 1 - E|)$ |
| Exponent Max | $E \cdot C_{\text{MUX}}$ |
| Mantissa MUL | $(m + 3)^2 \cdot C_{\text{AND}} + (m + 2)^2 \cdot C_{\text{FA}} + (m + 2) \cdot C_{\text{HA}}$ |
| Sort Exponent | $(M + 1) \cdot C_{\text{MUX}}$ |
| 1st Shift ($M + 1 >> k \rightarrow F$) | $(F - 1) \cdot \log_2(k_{\max}) \cdot C_{\text{MUX}}$ |
| Mantissa Adder ($F, F \rightarrow F$) | $(M) \cdot C_{\text{FA}} + C_{\text{HA}}$ |
| Leading Zero Detector | $F(C_{\text{AND}} + C_{\text{OR}}) + \log_2(k_{\max})^2 C_{\text{OR}}$ |
| 2nd Shift ($F >> k \rightarrow M + 1$) | $(M + 1) \cdot \log_2(k_{\max}) \cdot C_{\text{MUX}} - k_{\max} \cdot (C_{\text{FA}} - C_{\text{AND}})$ |
| Exponent Rebase | $(E - 1) \cdot C_{\text{FA}} + C_{\text{HA}}$ |
| Final Incrementor | $(M + 1)C_{\text{HA}}$ |

Table 9: FMA components gate-count breakdown. For the gate count, we used $C_{\text{AND}} = C_{\text{OR}} = 1$ for the standard gates AND2/OR2, $C_{\text{MUX}} = 3$ for MUX2, and $C_{\text{HA}} = 3, C_{\text{FA}} = 7$ for half and full adder.

We do not include Flip-Flops in our gate count. For the value of $F$ (Canvas bits, after shifting to fixed-point representation), we used $2M + 1$, the maximum bit width in which two 2's complementary values with $M + 1$ bits can interact during addition. For $k_{\max}$ (the maximum shift distance), we used $\min(\log_2(F), E)$, as the magnitude of the shift is bounded by both the number of exponent bits and the size of the canvas $F$.

| Weights/Activations bits | | FMAq Bits | | Canvas | | Gates | |
|---|---|---|---|---|---|---|---|
| m | e | M | E | F | $\log_2(k_{\max})$ | Count | Ratio [%] |
| 4 | 3 | 23 | 8 | 47 | 6 | 2208 | 100 |
| 4 | 3 | 10 | 5 | 21 | 5 | 1082 | 49 |
| 4 | 3 | 7 | 4 | 15 | 4 | 808 | 37 |

Table 10: Gate estimation for Quantized FMA

We summarize our numerical results in table 10. Our results show that for 8bit activations and weights (at FP format M4E3), as we used in the paper, any half-precision FMAq that follows our quantization

scheme is expected to reduce the number of gates by about 50% from the gate count of full-precision accumulators. Reducing the accumulator to M7E4, as was done in the paper, will cut the number of gates by an additional 25%, compared to half precision.

We conclude our 12bit accumulators will reduce the gate count by 63% compared to 32 bit FP accumulation. We note that the 16-bit accumulation gate count in our analysis is not directly applicable to previous works that used 16-bits accumulators– This is because in Sun et al. (2019), only the output of the accumulator was quantized with no explicit quantization of the internals. Presumably, the majority of the gain there was achieved by the reduction of communication bandwidth between FMAq components, which does not affect the gate count in this analysis.

## F    WHY IS IT HARD TO TRAIN WITH UNDERFLOW?

Our claim that underflow cause unique problems during SGD optimization is based, first and foremost, on empirical observations (see: Tab. 2, Tab. 3). In addition, we suggest several explanations to why this problem may occur when underflow is introduced during training, despite it having a small effect on the loss landscape.

Consider a neural network, where a specific scalar weight $w$ is connected to its scalar activation $x$ as input, and their product is $z = xw$. Suppose $w$ is small enough so that $z$ consistently results in product underflow, i.e. $Q_{\text{prod}}(z) = 0$. In this case, during forward propagation, the value of $x$ has little to no direct effect on the output neuron to which $w$ is connected. Therefore, it is reasonable to assume that the computed neural gradient $g \equiv \frac{d\mathcal{L}}{dz}$ (where $\mathcal{L}$ is the loss) will be uncorrelated with $x$. Consequently, the gradient update of the weight $w$ will be $\Delta_w \propto gx$, with the expected value $\mathbb{E}[\Delta_w] \propto \mathbb{E}[gx] = \mathbb{E}[g]\mathbb{E}[x]$. Based on previous quantization literature Banner et al. (2018), we have approximately $E[g] = 0$, and so $\mathbb{E}[\Delta_w] = 0$. Therefore, any sufficiently small weight $w$ will become "stuck", so that its $z$ cannot escape underflow for a long time.

The issue is excavated by the ratio between updates magnitude, and the magnitude a weight has to be updated to surpass the underflow threshold. In a fully-trained model, the gradients are expected to be $\frac{d\mathcal{L}}{dW} \simeq 0$. When transitioning to an LBA-model, we make sure to avoid significant changes to the loss landscape (as indicated by the zero-shot accuracy). As a result, we can expect the relative change in gradient to remain small, $|\Delta_w| = |\eta \frac{d\mathcal{L}}{dw}| \sim |w|$. (Otherwise, the loss landscape would change rapidly during SGD, and we can no longer consider the process as fine-tuning).

When dealing with quantized values, it is always possible that a gradient step will be too small to change the value. (This is the main motivation behind stochastic rounding, which is not suitable for our case). For example, for floating point quantization without underflow/overflow, the gradient step must be approximately $|\Delta_w| = |\eta \frac{d\mathcal{L}}{dw}| \geq 2^{-M}|w|$ for the quantized value of $w$ to 'jump' quantization level. In this case, $|\Delta_w| \sim |w|$ means that the ability of all weights to change during fine-tuning only depends on $M$, and the learning rate.

In the case of underflow, however, values must surpass an absolute threshold ($2^{-b}$), for the gradient step to have any effect. Consequently, under previous assumptions, any small enough value subjected to floating point quantization is expected to receive updates which are too small to result a state-change. This is what we referred to when mentioning values being 'stuck' and 'escaping'.

In LBA networks, the quantization is performed over intermediate values (products and partial accumulation values). These value do not get the explicit updates, but they will still get implicit updates by passing their respective neural gradient backwards.

