# OpenReview forum: "Towards Cheaper Inference in Deep Networks with Lower Bit-Width Accumulators"
_ICLR.cc/2024/Conference — ICLR 2024 poster_

### Official Review · Reviewer_AQt3 · 2023-10-26

**Soundness:** 2 fair
**Presentation:** 3 good
**Contribution:** 2 fair
**Rating:** 6
**Confidence:** 4

**Summary:**

This paper studies quantization errors occurring in the accumulation phase of dot products. While many works have studied representation quantization in the past, the problem of accumulation rounding has largely been overlooked. Yet, others (Higham'83 & Sakr'19) who have studied this problem, have often focused on the problem of swamping, which causes an underflow in one of the two summands. In contrast, this work looks at the problem of overflow that occurs during large summations, when employing relatively low precision accumulators, e.g., FP16. Modeling accumulation as a sum of iid random variables, and invoking the central limit theorem, this paper formulates a prediction method for when such overflows can occur, and as a remedy, applies a scaling factor the the chunked summation to try and suppress the damage of said overflows. Extensive empirical results on several benchmarks are provided.

**Strengths:**

-- Good problem to tackle, rounding effects in accumulators are largely overlooked in a community that puts a lot of emphasis on using quantization to lower the cost of implementation.

-- Good presentation. Following concepts related to the accumulation occurring in a GEMM is often hard, due to the complexities of tensor cores and similar hardware used for DL inference and training. But the others do a good job pinpointing the problem.

-- Solid empirical results on a diverse set of benchmarks such as vision and language models.

**Weaknesses:**

-- The proposed solution described in Section 3 is too qualitative. For instance, the underflow region is described as a "hard" region to "escape" from. Can the "escape" and "hardness" be presented in a mathematical manner? Similar for later description of parameters being "stuck" and so on.

-- The method only applies to feedforward, what about accumulations occurring in the GEMMs of back-propagation?

-- Benchmarks employed are diverse, but relatively simple (ResNets and Berts). can the empirical results be augmented with Transformer models, such as GPTs?

**Questions:**

Please address the above questions. Furthermore, I am also curious if we can analyze if overflows can be allowed in a controlled manner such as to lower complexity further but maintain accuracy. Indeed, overflowing is essentially related to magnitude clipping. Recent works [1] have shown that clipping, if done properly, can significant improve the quality of quantization. Is this something we can investigate for this work?

[1] Sakr, Charbel, et al. "Optimal clipping and magnitude-aware differentiation for improved quantization-aware training." International Conference on Machine Learning. PMLR, 2022.

---

> ### Author Response · Authors · 2023-11-16
> **Response to Reviewer AQt3**
>
> > The proposed solution described in Section 3 is too qualitative. For instance, the underflow region is described as a "hard" region to "escape" from. Can the "escape" and "hardness" be presented in a mathematical manner? Similar for later description of parameters being "stuck" and so on.
>
> We agree this was not sufficiently clear. Consider a neural network, where a specific scalar weight $w$ is connected to its scalar activation $x$ as input, and their product is $z=xw$. Suppose $w$ is small enough so that $z$ consistently results in product underflow, i.e. $Q_{\mathrm{prod}}(z) = 0$. In this case,  during forward propagation, the value of $x$ has little to no direct effect on the output neuron to which $w$ is connected. Therefore, it is reasonable to assume that the computed neural gradient $g \equiv \frac{d\mathcal{L}}{dz} $ (where $\mathcal{L}$ is the loss) will be uncorrelated with $x$. Consequently, the gradient update of the weight $w$ will be $\Delta_{w} \propto gx$, with the expected value $\mathbb{E}[\Delta_{w}] \propto \mathbb{E}[g x] = \mathbb{E}[g] \mathbb{E}[x]$. Based on previous quantization literature [1], we have approximately $E[g]=0$, and so $\mathbb{E}[\Delta_{w}]= 0 $. Therefore, any sufficiently small weight $w$ will become ``stuck", so that its $z$ cannot escape underflow for a long time.
>
> We added a new section in Appendix (Section F), for the theoretical of this issue.
>
> [1] Banner, Ron, et al. "Scalable methods for 8-bit training of neural networks." Advances in neural information processing systems 31 (2018).
>
> > The method only applies to feedforward, what about accumulations occurring in the GEMMs of back-propagation?
>
> True. Training with Low Bit width Accumulators (LBAs) is completely outside the scope of this paper, as we explain next. Efficient training would require LBAs in all phases of the backprop algorithm: forward propagation, backward propagation, and weight update. The last two phases cannot be directly fine-tuned, which was our approach for forward propagation in this paper. As a side note, we did experiment with training of neural networks with LBAs and were mostly unsuccessful. Therefore, at this point in time, we recommend LBA for inference only.
>
> > Benchmarks employed are diverse, but relatively simple (ResNets and Berts). can the empirical results be augmented with Transformer models, such as GPTs?
>
> Please note Table 5: the results there are on the Lamma2-7B model (i.e., which has 7 billion parameters). This is as close to GPT as we can get given our experiments rely on simulated hardware for the LBA (which is much less efficient than standard training).
>
>  Also note, that this model introduces a new difficulty since the range of values in the network varies greatly from layer to layer (which we attribute to splitting the layers to LoRA a and LoRA b). Therefore, a single exponent bias is not sufficient for using $M7E4$ accumulator. This issue can be solved using a dynamic exponent-bias (changes from layer to layer), and using this method, we were able to achieve $45.1$% validation MMLU score when fine-tuning Llama2 (baseline is $45.3$%), or $35.2$% when fine-tuning LLamaV1 (baseline $35.4$%). The results were added to the paper (Table 5), as well as a larger version of Bert in table (Table 5).
>
> > Furthermore, I am also curious if we can analyze if overflows can be allowed in a controlled manner such as to lower complexity further but maintain accuracy. Indeed, overflowing is essentially related to magnitude clipping. Recent works [1] have shown that clipping, if done properly, can significant improve the quality of quantization. Is this something we can investigate for this work?
>
> We have already seen evidence of this claim being true in our experiments. In some cases, representations with lower exponent count and carefully calibrated exponent-bias performed better than representations with a wider range. For example, zero-shot accuracy for $M7E5$ LBA on ResNets is lower than $M7E4$ with the optimal exponent-bias, despite having a lower representation range, which strongly indicates that a lower overflow threshold can be beneficial. We also attribute the small/negative gap between 3 and 4 exponent bits we saw in Table 7 (masked LM modelling) to this phenomenon.
>
> That being said, when we start fine-tuning models based on weights of non-LBA models, it is crucial that the model we start from is close enough to the pre-trained model. Excessive overflows will result in a substantial change in the function represented by the model (especially if the model is deep), to the point that we might as well start optimizing from scratch.

---

> > ### Comment · Reviewer_AQt3 · 2023-11-20
> > **Thanks for the response**
> >
> > I acknowledge having read the response by the authors. I am keeping my original review and score.

---

### Official Review · Reviewer_YwrL · 2023-10-30

**Soundness:** 3 good
**Presentation:** 3 good
**Contribution:** 3 good
**Rating:** 6
**Confidence:** 4

**Summary:**

This paper explores how to reduce computing resource by quantize and reducing the bit width of neural network computing accumulators. Unlike most other works that mainly focus on the multiplier part of low bit width, this paper believes that the overhead of the accumulator cannot be ignored. Therefore, through optimization on floating-point representation, the ResNet series models can achieve fine-tuning and inference for the first time with a lower 12 bit width accumulator, without significant degradation in accuracy. This paper also explores the training methods for lower bit width scenarios, including adjusting the design selection of the backpropagation straight through estimator.

**Strengths:**

1. Quantization optimization of accumulators is an interesting viewpoint, as it has not been given sufficient attention.

2. The method proposed in this paper has certain reference significance for the design of floating-point accumulators in deep learning accelerators.

3. The discuss of STE used in section 4 seems to be a new and original study, can achieve better results in backpropagation of accumulated errors.

**Weaknesses:**

1. The method proposed in this paper is difficult to be applied to accelerate real-world low bit width neural networks. Although this paper claims that an accumulator with a low bit width of 12 bits FP can be implemented with cheaper hardware, standard hardware typically only provides floating-point bit widths of 8 bits, 16 bits, and 32 bits. Implementing such acceleration requires specialized hardware design, which requires more collaborative design and additional costs. The optimization of low bit floating-point accumulators under the same cost, as well as the accuracy that can be achieved by fixed-point quantization models with the same hardware cost, remains to be discussed.

2. This paper lacks accurate evaluation and theoretical analysis of the error and accuracy requirements of floating-point models. Although Table 1 in Paper categorizes the errors in several cases of floating-point quantization, and Figure 2 shows several cases of errors, it is still not possible to quantitatively evaluate the impact of low bit width floating-point accumulation on model inference errors. It is difficult to make people believe with certainty the scalability and reliability of this scheme. Also, in the last paragraph before section 3.1, the author chooses b_acc=b_prod-1/2 log_2(chunk size) as an offset does not seem to guarantee that overflow will not occur under any condition. Therefore, I think that more discussion and theoretical analysis are needed regarding parameter selection and error evaluation.

3. The content of this paper only includes an evaluation of the model accuracy and does not discuss or analyze the actual cost. For example, regarding the estimation of multiplier and accumulators’ power and silicon area, how much performance improvement or resource savings can be achieved through the optimization proposed in this paper. It is interesting to discuss the benefits of these optimizations in the context of the additional accuracy loss and design complexity required by the methods presented in this paper.

**Questions:**

1. In section 3.1, a two-staged fine-tuning is proposed, what are the references for selecting hyperparameters here(e.g, 10 epochs, learning rate)?
2. In section 4, one method is using STE as recursive way on FMA. This seems to require all FMA operations to be unfolded in sequence. As far as I know, the FMA gradient of deep learning training seems to be treated equally, otherwise it will seriously affect efficiency and be unreality. This is because GEMM is a highly optimized parallel operation, and additional branches are not suitable for use here.

---

> ### Author Response · Authors · 2023-11-16
> **Response to Reviewer YwrL 1/2**
>
> > The method proposed in this paper is difficult to be applied to accelerate real-world low bit width neural networks. Although this paper claims that an accumulator with a low bit width of 12 bits FP can be implemented with cheaper hardware, standard hardware typically only provides floating-point bit widths of 8 bits, 16 bits, and 32 bits. Implementing such acceleration requires specialized hardware design, which requires more collaborative design and additional costs. The optimization of low bit floating-point accumulators under the same cost, as well as the accuracy that can be achieved by fixed-point quantization models with the same hardware cost, remains to be discussed.
>
> We respectfully disagree with how the reviewer classified numerical formats as "standard" vs "non-standard", in the context of deep learning accelerators. After all, the FP16 format (half precision, IEEE 2008), which is the default setup for deep learning applications, only had a bare-bone implementation in hardware before being heavily utilized in the Volta microarchitecture (2017). The FP8 formats (M2E5 /M3E4/ M4E3) do not have an IEEE definition and are not yet supported in standard hardware (Specifications were only published last year[1]). Our work is focused on a primary component of cutting-edge hardware AI accelerators. We believe the fact that FP12 is not standardized yet is a rather small obstacle in this context.
>
> Furthermore, if any hardware vendor does consider FP12 to be problematic, it can still make use of applying our method for half-precision accumulators. Half-precision accumulators do exist in the market, but they are not commonly used. We do not know the hardware design of existing products, but, as we discussed in the paper and the hardware analysis, our implementation does have clear potential to be more efficient than the FP16 LBA accelerators previously discussed in the literature, while providing more accurate models.
>
> [1] https://developer.nvidia.com/blog/nvidia-arm-and-intel-publish-fp8-specification-for-standardization-as-an-interchange-format-for-ai/
>
> > in the last paragraph before section 3.1, the author chooses $b_{\text{acc}}=b_{\text{prod}}-\frac12 \log_2(\texttt{chunk size})$ as an offset does not seem to guarantee that overflow will not occur under any condition.
>
> To be clear, the equation $b_{\text{acc}}=b_{\text{prod}}-\frac12 \log_2(\texttt{chunk size})$ only concerns the relative difference between the exponent bias terms. This is a relatively minor change, which is not intended or expected to prevent overflows on its own. To prevent overflows, the absolute value of $b_\text{acc}$ should be chosen using either layer-output statistics or zero-shot evaluation of pre-trained networks.
>
> >  This paper lacks accurate evaluation and theoretical analysis of the error and accuracy requirements of floating-point models. Although Table 1 in Paper categorizes the errors in several cases of floating-point quantization, and Figure 2 shows several cases of errors, it is still not possible to quantitatively evaluate the impact of low bit width floating-point accumulation on model inference errors. It is difficult to make people believe with certainty the scalability and reliability of this scheme... Therefore, I think that more discussion and theoretical analysis are needed regarding parameter selection and error evaluation.
>
> We agree with the reviewer's main point, that the effect of floating-point accumulation on models is not sufficiently clear from the paper, and we thank the reviewer for pointing this out. We added the following table (see: comment) to the appendix, describing the zero-shot accuracies of pre-trained models (ResNet, in this case) with different types of accumulator quantization. While zero-shot accuracies are not fully indicative of the quality of the fine-tuned LBA network, the table does showcase the challenge of reducing the bit-width, and the motivation behind the settings we had selected (M=7, E=4, $b_{\text{acc}}=10$). The new section, with this table, was added as Appendix B.
>
> > The content of this paper only includes an evaluation of the model accuracy and does not discuss or analyze the actual cost. For example, regarding the estimation of multiplier and accumulators’ power and silicon area, how much performance improvement or resource savings can be achieved through the optimization proposed in this paper. It is interesting to discuss the benefits of these optimizations in the context of the additional accuracy loss and design complexity required by the methods presented in this paper.
>
> See the general comment for our hardware analysis. The quantization scheme in this paper was made with chip-design limitations in mind, and should not result in any additional design complexity. For example, this is the reason we insisted on using bit-masking (no additional logic), and avoided more accurate yet expensive methods, like nearest-neighbour-rounding.

---

> > ### Author Response · Authors · 2023-11-16
> > **Table: Zeroshot accuracies for ResNet**
> >
> > | Mantissa Effect             |          |       |                |        |        |                                        |
> > |-----------------------------|----------|-------|----------------|--------|--------|----------------------------------------|
> > | Model                       | Baseline | M10E5 | M9E5           | M8E5   | M7E5   | M6E5                                   |
> > | ResNet18                    | 69.75    | 69.50 | 68.95          | 66.70  | 57.09  | 20.49                                  |
> > | ResNet34                    | 73.31    | 73.17 | 72.68          | 70.46  | 60.07  | 17.19                                  |
> > | ResNet50                    | 76.12    | 75.95 | 75.57          | 73.70  | 64.94  | 19.48                                  |
> > | **Exponent Bias Effect (M7E4)** |          |       |                |        |        |                                        |
> > | Model                       | $b=8$    | $b=9$ | $b=10$         | $b=11$ | $b=12$ | $b_{\text{acc}},b_{\text{prod}}=10,12$ |
> > | ResNet18                    | 55.68    | 60.64 | 60.00          | 58.84  | 56.96  | **60.14**                         |
> > | ResNet34                    | 50.80    | 63.30 | **63.88** | 62.46  | 59.90  | 63.65                                  |
> > | ResNet50                    | 26.41    | 64.25 | **68.69** | 67.57  | 66.12  | 68.49                                  |

---

> ### Author Response · Authors · 2023-11-16
> **Response to Reviewer YwrL 2/2 (Questions)**
>
> > In section 3.1, a two-staged fine-tuning is proposed, what are the references for selecting hyperparameters here(e.g, 10 epochs, learning rate)?
>
> In addition to hyperparameters used in this experiment, we have also run a set of experiments using fixed-learning rates (no cosine annealing). In the other set, we tested a few initial learning rate values (1E-7, 3E-8, 1E-8) for a few epochs, and used enough epochs to reach convergence (for training accuracy/loss) with the best-performing learning rate. The results in this regime were slightly better than the results published in the paper: For 8bit quantized ResNets with 4ME3, we achieved 69.6\% for Resnet18, 73.48\% for ResNet34 and 76.35\% for ResNet50.  However, this required more epochs and finer-tuned hyperparameters (different models used different learning rates). In the paper, we used the regime with cosine annealing since it was more robust to hyperparameter changes. We used $10$ epochs in the 1-stage compared with $5$ epochs for the dual-stage to support our claims that the gaps between the methods (where they exist) are not simply a result of better hyperparameters. The epoch count was initially chosen due to time-constraints and was kept since the benefit of running more epochs was small. We added his explanation to the Implementation details, in Appendix C.
>
> > In section 4, one method is using STE as recursive way on FMA. This seems to require all FMA operations to be unfolded in sequence. As far as I know, the FMA gradient of deep learning training seems to be treated equally, otherwise it will seriously affect efficiency and be unreality. This is because GEMM is a highly optimized parallel operation, and additional branches are not suitable for use here.
>
> The reviewer is correct about GEMM being highly parallelized, and our experiments were designed with this in mind. However, it is not true that the recursive STE requires the operation to be unfolded. All simulated GEMM operations and gradient estimation in this paper used a chunk size of $16$, which means that an operation with an accumulation width of $N$ is initiated with $\frac{N}{16}$ parallel operations (i.e., the first hierarchy), before aggregating the results (i.e., the second hierarchy). In the case of recursive STE, every detection of OF/DIFF during re-computation will result in a `$0$' in all preceding operations, just as we saw for sequential accumulation. The only difference for parallel accumulation is that the hierarchy tree can expand in two directions (Like Figure 1 (right) in the paper, with all the arrows reversed). We added this explanation to Appendix D (Old Appendix C, where we discussed fine grained methods in details)

---

> > ### Comment · Reviewer_YwrL · 2023-11-17
> >
> > Thank you for your detailed response and explanation. The revised appendix of the paper covers most of my concerns and makes the method more clear.
> >
> > I will change the rating from 3 to 6, while also increasing the soundness and contribution ratings.

---

### Official Review · Reviewer_AH1V · 2023-11-01

**Soundness:** 2 fair
**Presentation:** 3 good
**Contribution:** 3 good
**Rating:** 5
**Confidence:** 3

**Summary:**

In addition to low-precision inputs to a matmul, this paper proposes to use low-bitwidth accumulators (LBA) to compute the dot product. Experiments show that 12-bit LBA in the forward pass is promising for both ResNet and BERT, requiring only several epochs of finetuing. LBA for training is more difficult.

**Strengths:**

1. Reducing the accumulator bitwidth is a practical way of further reducing the cost of a low-precision matmul.

2. The experimental design in the main experiments in Section 3.1 is straightforward and seems to be easy to reproduce (if the CUDA kernel is open sourced).

**Weaknesses:**

1. The paper needs to provide more context on related works. For example, what is the key difference between the proposed method and the prior work Wrapnet? Is it training vs. non-training?

2. There is a lack of evaluation or estimation on the hardware benefits of the proposed method. What will be the gate count/computational energy/latency improvement if using 12-bit accumulators compared to FP16/BF16 accumulators?

3. Low-bitwidth accumulator under the context of integer quantization is not explored in the main experiments. Integer quantization is mentioned in Section 2.2 when introducing fixed-point quantization, but it seems to be disconnected from the rest of the evaluation.

**Questions:**

Questions are included in the weakness section.

---

> ### Author Response · Authors · 2023-11-16
> **Response to reviewer AH1V**
>
> > The paper needs to provide more context on related works. For example, what is the key difference between the proposed method and the prior work Wrapnet? Is it training vs. non-training?
>
> WrapNet, like our work, is also directed toward inference. The main difference in the setting between our work and WrapNet is that WrapNet uses Integer quantization, while our work uses floating point quantization and arithmetic. The integer quantization of the accumulator results in different challenges, as there, swamping is not an issue, and overflow becomes the main problem. (Presumably, this is why using very low bit weights seems to be mandatory to reduce overflow in WrapNet; but this also directly harms its final test accuracy). We expanded the related work discussion in section 2.2, and made sure this is better clarified in the last paragraph of 3.1, where we discuss the differences.
>
> > There is a lack of evaluation or estimation on the hardware benefits of the proposed method. What will be the gate count/computational energy/latency improvement if using 12-bit accumulators compared to FP16/BF16 accumulators?
>
> See the general comment for our hardware analysis. As we explained there, we could not reliably evaluate computation energy or area, but we did include a gate count estimation.
>
> >   Low-bitwidth accumulator under the context of integer quantization is not explored in the main experiments. Integer quantization is mentioned in Section 2.2 when introducing fixed-point quantization, but it seems to be disconnected from the rest of the evaluation.
>
> Correct. Our work is strictly focused on floating point quantization (as we explained in the introduction, we were motivated to do this by the recent popular trend of floating point quantization in transformers [1]). Integer accumulators were already explored in prior work (WrapNet). Fixed-point quantization was mentioned in the preliminary section to give full context and as a warm-up for the less trivial, floating-point quantization
>
> [1] https://docs.nvidia.com/deeplearning/transformer-engine/user-guide/examples/fp8_primer.html

---

> ### Author Response · Authors · 2023-11-21
> **Response To Reviewer AH1V: End of discussion period**
>
> We again thank you for your time and valuable feedback.
> Having just two days left for the discussion period, we would love to address any remaining issues, if there are any. If not, please consider increasing our score.

---

### Official Review · Reviewer_L15P · 2023-11-06

**Soundness:** 4 excellent
**Presentation:** 3 good
**Contribution:** 3 good
**Rating:** 6
**Confidence:** 4

**Summary:**

The paper introduces an innovative approach to fine-tune the process of quantized accumulation by disregarding underflow effects, thereby simplifying the rounding process to a simple floor operation during low-precision accumulation. Employing this technique, deep learning inference with 12-bit floating-point accumulation maintains the same as FP32 accumulation on the ImageNet dataset. Additionally, the paper suggests a methodology for quantized backpropagation across the entire accumulation, demonstrating promising results with 8-bit floating-point accumulation on smaller datasets, such as MNIST.

**Strengths:**

1- The paper is well-written and well-organized

2- The efficacy of the methods presented in the paper is substantiated through experimental results using BERT models on the SQuAD benchmark and ResNet models on ImageNet.

3- The backpropagation through the entire quantized accumulation is unique and has not been studied before.

**Weaknesses:**

1- It is recommended that the paper include a comparison of the computational complexity of MAC operations between the proposed method and the previous works [1,2,3]. Additionally, the author suggested discussing the quantization overhead associated with the new approach in comparison to [1,2,3].

[1] Sun, Xiao, et al. "Hybrid 8-bit floating point (HFP8) training and inference for deep neural networks." Advances in neural information processing systems 32 (2019).

[2] Sun, Xiao, et al. "Ultra-low precision 4-bit training of deep neural networks." Advances in Neural Information Processing Systems 33 (2020): 1796-1807.

[3] Chmiel, Brian, et al. "Logarithmic unbiased quantization: Simple 4-bit training in deep learning." arXiv preprint arXiv:2112.10769 (2021).

2- The impact of chunk size requires further exploration to determine if reducing the chunk size also diminishes the bit size.

**Questions:**

Can the distribution of the accumulations (which might follow a normal or other distribution) affect the performance of the quantization approach?

---

> ### Author Response · Authors · 2023-11-16
> **Response to Reviewer L15P**
>
> > It is recommended that the paper include a comparison of the computational complexity of MAC operations between the proposed method and the previous works [1,2,3].
>
> See the general comment for our hardware analysis. As for [2,3] (4-bit quantization), we did not include a direct comparison with 4-bit neural networks, since the changes are orthogonal, and we fully expect 4-bit neural networks to work with LBA once their implementation is standardized. From our experience, 4-bit can be adapted to residual neural networks with some effort, but is still too experimental for modern transformers (especially for activations).
>
> > Additionally, the author suggested discussing the quantization overhead associated with the new approach in comparison to [1,2,3].
>
> There is no explicit quantization overhead in our method. Since we rely on bit-masks (with no rounding), our method is similar to just removing logic connections within the FMA design, which should always translate to cheaper hardware post-synthesise. We added a small clarification for this in section 3, where we present the FMAq component.
>
> > The impact of chunk size requires further exploration to determine if reducing the chunk size also diminishes the bit size.
>
> Typically, higher chunk-size reduces the error in floating point accumulation [4] and was shown to do so in neural networks as well [1], up until the point where chunk-size $=\sqrt{N}$. In almost all experiments we have chunk-size $<\sqrt{N}$. The only exception is the case where LoRA is applied (like in our Table 5 experiment). There, some layers have a very low accumulation size, and the chunk-size can easily exceed $\sqrt{N}$.
> Generally, we aimed to avoid treating the chunk-size as a configurable hyperparameter, since in practice, the chunk-size used during GEMM operations is a result of the accelerator architecture. This is highlighted in section 2.4, where we discuss the previous works, including [1].
>
> > Can the distribution of the accumulations (which might follow a normal or other distribution) affect the performance of the quantization approach?
>
> It is always true that the ideal quantization approach (e.g., in terms of MSE), will change depending on the statistics of the quantized values. Due to the large range of represented values and the low relative error, floating-point quantization is relatively robust, but it is expected to be sub-optimal in cases where the distribution of values requires a smaller range.
>
> [1] Sun, Xiao, et al. "Hybrid 8-bit floating point (HFP8) training and inference for deep neural networks." Advances in neural information processing systems 32 (2019).
>
> [2] Sun, Xiao, et al. "Ultra-low precision 4-bit training of deep neural networks." Advances in Neural Information Processing Systems 33 (2020): 1796-1807.
>
> [3] Chmiel, Brian, et al. "Logarithmic unbiased quantization: Simple 4-bit training in deep learning." arXiv preprint arXiv:2112.10769 (2021).
>
> [4] Higham NJ. The accuracy of floating point summation. SIAM Journal on Scientific Computing. 1993 Jul;14(4):783-99.
>
> [5] Ni, Renkun, et al. "Wrapnet: Neural net inference with ultra-low-resolution arithmetic." *arXiv preprint arXiv:2007.13242* (2020).

---

> > ### Comment · Reviewer_L15P · 2023-11-23
> > **Reviewer Response**
> >
> > I acknowledge having read the response by the authors. I maintain my initial score and review.

---

> ### Author Response · Authors · 2023-11-21
> **Response To Reviewer L15P: End of discussion period**
>
> We thank the reviewer again for the effort given, the recommendations for improving the paper and for recognizing the strengths in our work.
> Having just two days left for the discussion period, we would love to address any remaining issues, or further suggestions for strengthening the paper, if there are any. If not, please consider increasing our score.

---

### Author Response · Authors · 2023-11-16
**General Rebuttal**

We thank the reviewers for their constructive feedback on our work, and for appreciating the significance and novelty of our work. In the following, we address all the reviewer comments point-by-point.

Several reviewers commented on the lack of an in-depth hardware analysis in the paper. Therefore, we give an estimate of the number of logic gates. We do this for different variations of the quantized Fused Multiply Accumulator (FMAq). The details of how this was done are attached in the comment to this message.

We summarize our results in the following table. The results show that for 8-bit activations and weights (at FP format M4E3), as we used in the paper, and follow our quantization schemes. There, we see that 16-bit FMAq reduces the number of gates by about 50\%  from the gate count of full-precision 32-bit accumulators. Further reducing the accumulator to 12-bit (M7E4, as was done in the paper) will cut the number of gates by an additional 25\%, compared to 16-bit.

| Weights/Activations | FMAq bits | Canvas (F) | Gates | Ratio [%] |
|---------------------|-----------|------------|-------|-----------|
| M4E3                | fp32      | 47         | 2208  | 100       |
| M4E3                | fp16      | 21         | 1082  | 49        |
| M4E3                | M7E4      | 16         | 808   | 37        |

**We conclude our 12bit accumulators will reduce the gate count by $63$% compared to $32$ bit FP accumulation.**
 Please note the $16$-bit accumulation gate count in our analysis is not directly applicable to previous work that used 16-bit accumulators [1]. This is because in [1], only the output of the accumulator was quantized with no explicit quantization of the internals. Presumably, the majority of the gain there was achieved by the reduction of communication bandwidth between FMAq components, which does not affect the gate count in this analysis.

We thank the reviewers for highlighting the necessity of a hardware analysis, which was added to the paper (Appendix E). Of course, gate-count analysis does not give a full picture of the hardware cost. Accurately finding the amount of power/area saved by applying lower-bit width accumulators at different bit-widths requires synthesizing the suggested hardware component. However, we do not have the tools for such a synthesis. Moreover, it is a common practice in the deep learning acceleration literature that algorithms appear before the hardware that can support them --- since without a working algorithm, there is no reason to invest in hardware to support new functionality.

[1] Sun, Xiao, et al. ``Hybrid 8-bit floating point (HFP8) training and inference for deep neural networks." Advances in neural information processing systems 32 (2019).

---

> ### Author Response · Authors · 2023-11-16
> **Gate count estimate technical details**
>
> Following an existing design of FMAq component [2,figure 2b], we adjusted the design for the case of FMAq with m/e quantization of weights and activations and M/E quantization of intermediate values (product, accumulator), and suggested the following gate counts, as seen in the following table. The gate counts are all based on the gate count assumptions listed in [2, appendix B], and common block designs.
>
> | FMA Components breakdown     | Gate Count                                                                                       |
> |------------------------------|--------------------------------------------------------------------------------------------------|
> | Exponent Adder               | $(e-1)\cdot C_\text{FA}+ C_\text{HA}$                                                            |
> | Exponent Differ              | $(\min(E,e+1)-1)\cdot C_\text{FA}+ C_\text{HA} \cdot (1+\|e+1-E\|)$                              |
> | Exponent Max                 | $E\cdot C_\text{MUX}$                                                                            |
> | Mantissa MUL                 | $(m+3)^2 \cdot C_\text{AND}+(m+2)^2 \cdot C_\text{FA}+(m+2) \cdot C_\text{HA}$                   |
> | Sort Exponent                | $(M+1)\cdot C_\text{MUX} $                                                                       |
> | 1st Shift ($M+1 >> k \to F$  | $(F-1) \cdot \log_2 (k_{\max}) \cdot C_\text{MUX} $                                              |
> | Mantissa Adder ($F,F\to F$)  | $(M)\cdot C_\text{FA}+ C_\text{HA}$                                                              |
> | Leading Zero Detector        | $F(C_\text{AND}  + C_\text{OR})+\log_{2} (k_{\max})^2 C_\text{OR}$                               |
> | 2nd Shift ($F >> k \to M+1$) | $(M+1) \cdot \log_{2} (k_{\max})  \cdot C_\text{MUX} -k_{\max} \cdot (C_\text{FA}-C_\text{AND})$ |
> | Exponent Rebase              | $(E-1)\cdot C_\text{FA}+ C_\text{HA}$                                                            |
> | Final Incrementor            | $(M+1) C_\text{HA}$                                                                              |
>
> The table details our FMA components gate-count breakdown. For the gate count, we used $C_\text{AND}=C_\text{OR}=1$ for the standard gates AND2/OR2, $C_\text{MUX}=3$ for MUX2,  and $C_\text{HA}=3, C_\text{FA}=7$ for half and full adder.
>
> We do not include Flip-Flops in our gate count. For the value of $F$ (Canvas bits, after shifting to fixed-point representation), we used $2M+1$, the maximum bit width in which two 2's complementary values with $M+1$ bits can interact during addition. For $k_\max$ (the maximum shift distance), we used $\min(\log_2(F),E)$, as the magnitude of the shift is bounded by both the number of exponent bits and the size of the canvas $F$.
>
> [2] van Baalen, Mart, et al. ``FP8 versus INT8 for efficient deep learning inference." *arXiv preprint arXiv:2303.17951* (2023).

---

### Meta-Review · Area_Chair_9hnb · 2023-12-11

**Metareview:**

This paper proposes a method to accelerate neural network inference by reducing the accumulator bit-width. They reduce the accumulator precision to 12-bit for ResNets and language models, with no major accuracy degradation. They provide a gate count result to justify the benefit of reducing accumulator bit-width. Reducing accumulator bit-width is a problem worth-studying, and the paper has some novelty and contributions towards that goal. Weaknesses include that the benefit of low-bit accumulators are not fully shown, and error analysis could be improved.

**Justification For Why Not Higher Score:**

the benefit of low-bit accumulators are not fully shown, and error analysis could be improved.

**Justification For Why Not Lower Score:**

Reducing accumulator bit-width is a problem worth-studying, and the paper has some novelty and contributions towards that goal.

---

### Decision · Program_Chairs · 2024-01-16

Accept (poster)